# Structural and Functional Characterization of Stx2k, a New Subtype of Shiga Toxin 2

**DOI:** 10.3390/microorganisms8010004

**Published:** 2019-12-18

**Authors:** Anna C. Hughes, Yuzhu Zhang, Xiangning Bai, Yanwen Xiong, Yan Wang, Xi Yang, Qingping Xu, Xiaohua He

**Affiliations:** 1Western Regional Research Center, United States Department of Agriculture, Agricultural Research Service, 800 Buchanan St., Albany, CA 94710, USA; anna.hughes@usda.gov (A.C.H.); yuzhu.zhang@usda.gov (Y.Z.); 2State Key Laboratory for Infectious Disease Prevention and Control, National Institute for Communicable Disease Control and Prevention, Chinese Center for Disease Control and Prevention, P. O. Box 5, Changping, Beijing 102206, China; baixiangning@icdc.cn (X.B.); xiongyanwen@icdc.cn (Y.X.); wangyan@icdc.cn (Y.W.); yangxi_163cdc@163.com (X.Y.); 3GM/CA, Advanced Photon Source, Argonne National Laboratory, 9700 S. Cass Ave., Lemont, IL 60439, USA; qxu@anl.gov

**Keywords:** crystal structure, cytotoxicity, Gb3 and Gb4 receptors, immunoassays, Shiga toxin, neutralization, STEC, toxin stability

## Abstract

Shiga toxin (Stx) is the major virulence factor of Shiga toxin-producing *Escherichia coli* (STEC). Stx evolves rapidly and, as such, new subtypes continue to emerge that challenge the efficacy of existing disease management and surveillance strategies. A new subtype, Stx2k, was recently identified in *E. coli* isolated from a wide range of sources including diarrheal patients, animals, and raw meats, and was poorly detected by existing immunoassays. In this study, the structure of Stx2kE167Q was determined at 2.29 Å resolution and the conservation of structure with Stx2a was revealed. A novel polyclonal antibody capable of neutralizing Stx2k and an immunoassay, with a 10-fold increase in sensitivity compared to assays using extant antibodies, were developed. Stx2k is less toxic than Stx2a in Vero cell assays but is similar to Stx2a in receptor-binding preference, thermostability, and acid tolerance. Although Stx2k does not appear to be as potent as Stx2a to Vero cells, the wide distribution and blended virulence profiles of the Stx2k-producing strains suggest that horizontal gene transfer through Stx2k-converting phages could result in the emergence of new and highly virulent pathogens. This study provides useful information and tools for early detection and control of Stx2k-producing *E. coli*, which could reduce public risk of infection by less-known STECs.

## 1. Introduction

Infection with Shiga toxin-producing *Escherichia coli* (STEC) can manifest in a range of symptoms from mild diarrhea to the potentially life-threatening hemolytic uremic syndrome (HUS) [1,2]. The digestive tract of ruminants is the predominant reservoir for STEC [3,4,5]. STEC infections can also arise from other contaminated livestock, like swine and other non-ruminants, as well as contaminated water or fresh produce [6,7,8,9]. Public health experts have estimated that there are approximately 265,000 STEC infections a year in the US [10]. In fact, at the time of this writing there are two ongoing multistate outbreaks in the US, resulting in 110 illnesses and 61 hospitalizations, which have been traced to contaminated lettuce and an unidentified ingredient in a chopped salad kit [11].

Shiga toxin (Stx) is the major virulence factor of STEC and is an AB_5_ toxin consisting of a single A-subunit associated with 5 B-subunits (AB_5_) [12,13]. The A-subunit encodes an active *N*-glycosidase that binds to the 60S ribosomal subunit and removes a conserved adenine residue from the 28S rRNA, which inhibits host cell translation and results in cell death [14,15]. The B-subunit is a receptor-binding protein. Binding to either the globotriose (Gb3)-lipopolysaccharide (LPS) or the globotetraose (Gb4)-LPS, glycolipids on the cell membrane mediate holotoxin entry into host cells through endocytosis (both clatherin dependent and independent) and ultimate retro-transport from the ER to the cytosol [16,17,18]. While all Stxs conform to the basic AB_5_ structure and act by inhibiting translation, STEC pathogenesis is highly dependent on the type(s) of Stx expressed by the organism [19,20].

Stx can be divided into two main types, Stx1 and Stx2. Stx1 is almost identical to Stx produced by *Shigella dysentariae*, for which it was named [21]. In contrast, the immunologically distinct type 2 (Stx2) shares 55% and 57% sequence identity to the Stx1 A- and B- subunits, respectively [22,23]. Stx2 is often more toxic than Stx1 and infections with *stx2*-harboring STEC are more likely to develop into HUS than infections with *stx1*-carrying STEC [13,24,25]. Both Stx1 and Stx2 target Gb3 receptors, however, it is thought that a slower Stx2a-receptor dissociation may contribute to a greater Stx2 toxicity [16]. Seven subtypes of Stx2 (Stx2a to Stx2g) that vary in cytotoxicity, receptor binding, severity of symptoms, and associated host reservoirs have been reported [19,26]. The subtypes of Stx2a, Stx2c, and Stx2d have been isolated from humans and are more frequently associated with the development of colitis and HUS [27,28]. STEC harboring Stx2b, Stx2e, Stx2f, and Stx2g are commonly isolated from non-human sources, such as deer, pigs, and cattle. Nonetheless, there are sporadic reports of human infections with STEC carrying less common Stx subtypes, suggesting that no subtype is confined to animal hosts [29,30]. Recently, two new subtypes of Stx2 were identified from STEC strains isolated from marmots (Stx2h) and shrimp (Stx2i) [31,32]. The increased occurrences of human infections with STEC strains harboring new Stx2 subtypes suggest that the less common subtypes may be of more clinical importance than previously thought.

Stxs are encoded on lambdoid prophages integrated into STEC genomes [33]. The Shiga toxin gene is encoded as a dicistron with the A-subunit coding gene proceeding the B-subunit gene, separated by a small intergenic sequence driven by a late phage promoter. Induction of the SOS response results in phage replication and toxin production, which are released upon bacterial lysis [34]. Toxin in the extracellular milieu can cross the host epithelial lining into the bloodstream, where it is transported to target sites, such as the kidneys and central nervous system [35]. Released phage particles can infect new *E. coli* hosts. Horizontal gene transfer of the prophage likely accounts for the diversity of serotypes and ancillary virulence factors associated with infection as well as the emergence of new subtypes [33,36].

Recently, we identified a new subtype of Stx2, designated as Stx2k, in strains isolated from a broad range of hosts in China, including diarrheal patients, goat, pig, and raw meat (beef and mutton) [7,37]. Stx2k shares 39.9% to 96.2% similarity at the nucleic acid level, and 68.1% to 97.0% similarity at the amino acid level with other subtypes of Stx2 [37]. Stx2k-producing strains were genetically heterogeneous in serotype, genome sequence, Stx2k-converting phage type, and virulence gene profile. Some strains even possess characteristics from both enterohemorrhagic *E. coli* (EHEC) and enterotoxigenic *E. coli* (ETEC), which suggests that Stx2k-converting phages may contribute to the rise of new virulent strains [37]. In this study, we generated a Stx2k recombinant toxoid and report the X-ray crystal structure of the Stx2k toxoid. All residues in the active-site are conserved between Stx2a and Stx2k as expected based on sequence similarity, but there are two amino acid differences in the receptor-binding site. A new antibody and an immunoassay were developed for sensitive and quantitative detection and neutralization of Stx2k produced by wild type STEC strains. The potential virulence of Stx2k was evaluated by comparing its biological properties including cytotoxicity, thermostability, acid tolerance, and receptor binding with the archetypical Stx2a subtype.

## 2. Materials and Methods

### 2.1. Cloning, Expression, and Purification of *Stx2kE167Q* Toxoid

Bacterial strains used in this study are listed in Table 1. To produce non-active Stx2kE167Q toxoid, the full-length *stx2kE167Q* gene (including coding sequences for the A- and B-subunits separated by the 11 bp intergenic region: aggagttaagt) with LIC fusion tags (5′TACTTCCAATCCAATGCA3′ at the N-terminus and 5′TTATCCACTTCCAATGTTATTA3′ at the C-terminus) was synthesized by IDT based on GenBank sequence (KC339670.2) [37]. The synthesized DNA was then cloned into the pET His6 TEV LIC vector-1B following instruction for the Addgene plasmid #29653 (https://www.addgene.org/29653/). The resulting construct was transformed into BL21 DE3 cells. For protein expression, an overnight culture (20 mL) was diluted 1:50 into 1 L of Lysogeny Broth (LB) medium supplemented with 50 µg/mL of kanamycin. The cell culture was then incubated at 37 °C with shaking until it reached an Optical Density of 0.6 at the wavelength 600nm (OD_600_); at this point, protein production was induced by addition of Isopropyl β- d-1-thiogalactopyranoside (IPTG) at a final concentration of 1 mM at 16 °C for 20 h. Afterwards, the cells were harvested by centrifugation at 8000 rpm for 30 min at 4 °C. The cell pellet was lysed in 1:10 volume of phosphate-buffered saline (PBS) by sonication. The lysate was clarified by centrifugation and precipitated with ammonium sulfate. The protein fraction precipitated from 40–60% ammonium sulfate was harvested by centrifugation at 12,000× *g* for 15 min at 4 °C and the pellet was rehydrated in PBS for 1 h at room temperature. After desalting using a Zeba Spin Desalting Column (7K MWCO, Thermo Scientific, Waltham, MA, USA), samples containing Stx2kE167Q were purified by affinity chromatography using an affinity column (Thermo Scientific AminoLink™ Plus Immobilization Kit, Waltham, MA, USA) coupled to the mAb Stx2e-3 [38], followed by gel filtration on an AKTA FPLC using Superdex 200-XK 26/70 column (GE Healthcare, Marlborough, MA, USA) as described previously [39,40]. The purified protein was concentrated and buffer exchanged as needed. Stx2kE167Q did not result in cytotoxicity to Vero cells (Vero cells incubated with 10 ng/mL of Stx2kE167Q were not significantly different from Vero cells incubated with the media only control).

### 2.2. E. coli Strains and Growth Conditions

Wild type bacterial strains were incubated overnight at 37 °C in LB with shaking and then diluted 1:50 into fresh LB medium supplemented with 100 ng/mL mitomycin C (MMC) and grown overnight at 37 °C. The supernatants of bacterial cultures were harvest by centrifugation and filter sterilized (0.22 µM PES membrane).

### 2.3. Wildtype *Stx2k* and *Stx2a* Protein Purification

Stx2a and Stx2k were purified from culture supernatants in the same manner as the toxoid in cleared lysates as described above, however, Stx2a was purified using an affinity column coupled to a mixture of mAbs (mAb Stx2-1, which binds the A-subunit, and mAb Stx2-5, which binds both the A- and B-subunits [44]), and Stx2k was purified using an affinity column coupled to mAb Stx2e-3 [38].

### 2.4. Production and Purification of Rabbit Polyclonal Antibodies Against *Stx2k*

The polyclonal antibody (pAb) raised against Stx2k toxoid was produced by Pacific Immunology Corp (Ramona, CA, USA). Briefly, the Stx2k toxoid was emulsified with either Complete Freund’s adjuvant (1st immunization), or incomplete adjuvant (2nd to 4th boosts) prior to immunization. The emulsion was injected to two rabbits, 13907 and 13908, at 3-week intervals (~300 µg toxoid was injected per rabbit at each time point). Toxoid injection did not result in cytotoxicity to either rabbit. Following the 3rd injection, bleeds were collected and evaluated for anti-antigen activity by ELISA. Antibodies from the first bleed of rabbit 13908 were purified by affinity chromatography on a Protein- A conjugated agarose column (Pierce Protein A IgG Purification Kit–Thermo Scientific, Waltham, MA, USA) and bound antibodies were eluted with 0.1 M glycine-HCl, pH 2.7. Protein concentrations were determined based on OD at A280nm measured with an Eppendorf BioSpectrometer (Hamburg, Germany).

### 2.5. Crystallization and X-Ray Data Collection

Crystallization screening was carried out at room temperature (20 °C) with the sitting drop vapor diffusion method. The screens were set up using a Formulatrix NT8 (Formulatrix, Bedford, MA, USA) drop setter. The conditions of the JCSG-plus, ProPlex HT, PACT premier, and PGA-LM HT kits from Molecular Dimensions (Molecular Dimensions Inc, Maumee, OH, USA) were used. Sitting drops 0.1 µL of protein solution mixed with 0.1 or 0.05 µL of crystallization solution were sealed against 0.1 mL of reservoir solutions in 96-well UVXPO plates (Molecular Dimensions, Holland, OH, USA) with a total of 768 starting drop conditions. Crystals with different shapes were obtained in the sitting drops with A1, A4, A5, and H9 of ProPlex; B4, B5, C8, and D1 of JCSG-plus; and G4–G8 of PGA screens. Only the original hit conditions containing cryoprotectant were chosen to be optimized. The crystals used for data collection were grown with the hanging drop vapor diffusion method. The drops were manually set up by mixing 1 µL of protein and 1 µL reservoir solutions. The concentrations of the protein solution 13, 19, 10, and 10 mg/mL were used for optimization by varying the precipitant and cryoprotectant concentrations of the reservoir solutions based on JCSG-B4, JCSG B5, ProPlex A1, and ProPlex A4, respectively. The data set used for structure refinement was collected with crystals grown with the reservoir solution containing 0.1 M HEPES, pH 7.5, 9.2% *w*/*v* PEG 8000, and 15% *v*/*v* ethylene glycol.

X-ray diffraction data were collected at LRL-CAT, 31ID-B beam line at APS, Argonne National Laboratory. Data were collected at 110 K with a 200 mm crystal-to-detector distance using a Pilatus3 S 6M detector. One thousand eight hundred 0.2° frames were collected with an exposure time of 0.24 s. The diffraction data were processed with the XDS [45] and the HKL2000 [46] suite of programs.

### 2.6. Structure Solution and Refinement

The structure was solved by the molecular-replacement method using the program Phaser [47,48]. The initial homology models for the A- and B-subunits were built with the program Chainsaw [49] using the subunit structures of the Stx2A (PDB:1r4p) as templates [13]. Structure refinement was carried out with Phenix-refine [50] alternated with semi-automatic model building and model improvement using coot [51].

The final structure was refined with data to 2.29 Å resolution and the final model was checked by Molprobity Validation [52] and PROCHECK [53]. Molecular graphics were prepared using the program Pymol (http://pymol.org/). The structure of Stx2k toxoid has been submitted to the Protein Data Bank under accession number 6U3U.

### 2.7. Vero Cell Cytotoxicity Assay

Stx cytotoxicity was determined using a Vero cell assay. Vero cells (100 µL/well) were seeded at a density of 0.5 × 10^5^ cells/mL in black, clear bottom 96-well plates (Corning) and allowed to adhere overnight at 37 °C and 5% CO_2_ in Dulbecco’s Modified Eagle Medium (DMEM) supplemented with 10% FBS and 1 × glutamax (Gibco, Life Technologies Corporation, Grand Island, NY, USA). After 24 h, cells were treated with 100 µL/well fresh DMEM containing Stx diluted 10-fold from 1 g/mL to 0.01 fg/mL for 48 h at 37 °C and 5% CO_2_. Cell viability was assessed using CELLTITER-GLO 2.0 ASSAY (Promega, Madison, WI, USA) according to the manufacturer’s instructions, except that the reagent was diluted 1:5 in PBS prior to use. Luminescence was measured on a Victor3 plate reader (Perkin Elmer, Waltham, MA, USA). The percent toxicity was calculated as (the average counts per second (CPS) of the DMEM only control − the CPS of the experimental sample)/average DMEM control ×100% and was plotted on a semi log scale. The linear portion of the dose response curve was plotted and used to estimate the concentration of Stx needed to reach 50% cytotoxicity (CD_50_). The percent survival was calculated as the average CPS of the experimental sample/the average CPS of the DMEM control ×100%. All treatments were performed in triplicate and repeated three times.

### 2.8. Neutralization of *Stx2k* Cytotoxicity In Vero Cells

Shiga toxin 2k (5 ng/mL) was incubated with 10 µg/mL of the indicated antibodies for 1 h at 37 °C prior to adding to Vero cells. Antibody-alone and toxin-alone controls were included. Cytotoxicity was calculated as described above and the relative cytotoxicity after neutralization was calculated by normalizing the toxicity of Stx2k without antibody as 100%. All treatments were performed in triplicate and repeated three times.

### 2.9. Thermal Stability

Shiga toxins were incubated at a range of temperatures for 1 h then placed on ice to cool. The heat-treated toxins were diluted into prewarmed DMEM at a final concentration of 5 ng/mL and incubated with Vero cells. Buffer-only controls were included. Cytotoxicity was calculated as described above. All treatments were performed in triplicate and repeated three times.

### 2.10. pH Stability

Shiga toxins were diluted to 500 ng/mL in 250 mM sodium acetate buffer adjusted to the indicated pH and incubated for 1 h at room temperature. The toxins were then diluted to 5 ng/mL into prewarmed DMEM and incubated with Vero cells. Buffer-only controls were included. All treatments were performed in triplicate and repeated three times.

### 2.11. Receptor Binding

The Gb3/Gb4-LPS receptor-binding assays were performed as previously described [39]. Briefly, formaldehyde-fixed *E. coli* cells either expressing Gb3-LPS, Gb4-LPS, or control *E. coli* cells (CWG308) were diluted to 0.1 OD_600_ in carbonate buffer (0.1 M NaCO_3_, pH 9.6) and 100 µL was added to the wells of a 96-well ELISA plate and dried overnight at 50 °C. Wells were then blocked with 300 µL/well of 5% non-fat dry milk (NFDM) in PBS with 0.05% Tween-20 (PBST) for 1 h at room temperature. Wells were washed twice with PBST and then incubated with 100 µL of pure toxin (50 ng/mL) for 1 h at room temperature. After washing six times with PBST, wells were incubated with 100 µL of mAb Stx2e-3 (100 ng/mL) [38] for 1 h at room temperature. After washing six times with PBST, wells were incubated with 100 µL of horse radish peroxide (HRP)-conjugated goat anti-mouse antibody (Promega, Madison, WI, USA) (1:5000) for 1 h at room temperature. Signals were developed with 100 µL/well of SuperSignal West Pico Chemiluminescent Substrate (Thermo Scientific, Waltham, MA, USA) and luminescence (CPS) was read on a Victor3 plate reader (Perkin Elmer, Waltham, MA, USA) for 0.1 s. All treatments were performed in triplicate and repeated three times.

### 2.12. ELISA

Sandwich ELISA’s were performed as previously described [54]. Briefly, 96-well black NUNC Maxisorp flat bottom plates (Thermo Scientific, Waltham, MA, USA) were coated with 100 µL/well of 1 g/mL of the capture antibody and incubated at 4 °C overnight. After overnight incubation the plates were washed two times in 0.02 M Tris buffered saline with 0.9% NaCl, pH 7.4, and 0.05% Tween-20 (TBST), blocked in 5% NFDM-TBST for an hour, and then incubated with samples in either PBS or LB at room temperature for 1 h. The plates were washed in TBST six times and incubated with the indicated detection antibody (100 ng/mL in NFDM-TBST). Goat anti-mouse HRP, or goat anti-rabbit HRP was added as a secondary antibody after washing six times in TBST. Plates were developed with 100 µL SuperSignal West Pico Chemiluminescent Substrate (Thermo Scientific, Waltham, MA, USA) and luminescence was read on a Victor3 plate reader (Perkin Elmer, Waltham, MA, USA) for 0.1 s. The limit of detection (LOD) was defined as the lowest toxin concentration at which the average ELISA reading was three standard deviations above the negative control. For direct ELISA’s, the plates were directly coated with 100 µL/well of 10 ng/mL of Stx2k toxin, followed by blocking, washing, and incubation with rabbit antisera at the indicated dilution and then goat-anti Rabbit-HRP as described above.

## 3. Results

### 3.1. The Structure of *Stx2kE167Q* Toxoid

Stx2k is a new subtype of Stx2 and little information is available about its structure or function relative to other Shiga toxin subtypes. Therefore, we wanted to purify the toxin and determine its structure relative to Stx2a. Considering the difficulty of purifying large quantities of Stx2k from wild type pathogenic bacteria, a recombinant construct encoding a non-toxic toxoid of Stx2k was generated by converting the glutamic acid at position 167 to glutamine (Stx2kE167Q). The E167Q mutation was chosen because E167 is known to be the active site residue for known ribosomal inactivating proteins (RIPs), including the Stxs, and because the E167Q mutation does not alter the structure of RIPs [38,44,54,55,56,57]. The toxoid was expressed in BL21 bacterial cells with an average yield of 2 mg/L of culture. As shown in Figure 1, the Stx2kE167Q toxoid purified is pure without noticeable contamination and the sizes of the A- and B-subunits were as expected on the stained sodium dodecyl sulfate-polyacrylamide gel electrophoresis (SDS-PAGE). The Stx2kE167Q was confirmed to be non-toxic in Vero cell assays (Vero cells incubated with 10 ng/mL of Stx2kE167Q were not significantly different from those incubated with the media only control). This toxoid preparation was used to determine the crystal structure of Stx2kE167Q.

The best crystal resulted in a data set of 2.29 Å resolution. Data processing revealed a monoclinic crystal system P2_1_ with unit-cell parameters a = 57.18, b = 157.02, c = 107.41 Å, and β = 94.61°. Molecular-replacement resulted in a solution with two biological units in the crystallographic asymmetric unit. Thus, the solvent content of the crystal was 61.34%, with a Matthews coefficient of 3.18 Å^3^/Da. Each biological unit is composed of one A-subunit and five B-subunits. The refined structure gave R/R_free_ values of 0.197/0.225 for all data to 2.29 Å (Appendix A and Figure 2). The root mean square derivatives (RMSD) from ideal small-molecule empirical values were 0.005 Å and 1.0 degrees for bonds and angles, respectively [58]. There were no outliers for bond length or bond angles based on "ideal" small molecule values [59]. There were no chirality outliers or planarity outliers. On the Ramachandran plot of the final refined structure, 97.01% were in the favored region and 2.83% were in the allowed region. Val59 in the A-subunit in the unit cell was the only outlier. For the A-subunit, residues 243–256 were flexible and could not be located in the electron density map in one of the biological units, and residues 244–257 of the A-subunit in the other biological unit could not be located. The final structure also included a Zn ion, 2 HEPES, 7 ethylene glycol, and 116 water molecules. The assignment of the zinc to the metal sites was based on X-ray fluorescence scans and anomalous X-ray diffraction. Structural differences between the two biological units were very small, except for the last five residues at the C-terminus of the A-subunit (Figure 2). The difference can be attributed to the crystal packing of the C-terminus of the A-subunit in one of the biological units. The RMSD for the C_α_ of all residues in the two biological units was 0.65 Å.

Sequence alignment indicates that Stx2k and Stx2a have 95% identity (Appendix A) [37]. As expected, based on sequence identity, the structures of Stx2kE167Q and Stx2a are similar (Figure 2B). The most notable similarity was the helix close to the C-terminus of the A-subunit, which was packed in the cavity at the center of the pentameric complex of the B-subunit (Figure 2B). The RMSD for C_α_ of superposing the structure of Stx2a with one of the biological units of Stx2k was 1.2 Å, and 1.4 Å with the other biological unit. Many of the non-identical residues between Stx2k and Stx2a were located on the surface of the structure (Figure 2C). While the differences may not affect the packing of the toxin, mutations on the surface of the protein may impact affinity to other molecules and could affect toxicity. Two of the four residues (AA16 and AA31) in the B-subunit that differ between Stx2a and Stx2k are involved in receptor binding (see below). Mutations on the surface may also affect antibody binding in immunoassays. One of the amino acid differences in the A-subunit is AA63, which is an Arg in Stx2a and a His in Stx2k. Together with H66, H63 coordinate a zinc ion at the interface of crystal packing among the different crystallographic units in Stx2kE167Q but not in Stx2a.

### 3.2. The Active Site

Amino acid residues critical for Stx activity (the active site) were identified by X-ray crystallography, sequence alignment with active sites of other RIPs, and assessing the toxicity of various Stx mutants in yeast [15,60]. Superimposing the active center residues of Stx2a and Stx2kE167Q revealed minimal changes in their spatial configuration, suggesting that any variation in toxin potency may be due to the internalization of the toxins dictated by the B-subunit, rather than the enzymatic activity of the A-subunit. In the crystal structure of Stx2k toxoid, there is a HEPES molecule in the active center of the A-subunit (Figure 3A). However, it is unclear if HEPES affects activity of the toxin.

### 3.3. Receptor-Binding Site

Residues important for receptor-interaction have been identified by structural studies of Stx2a in complex with its receptor or receptor analogues [61,62,63]. These residues constitute three sites for association with the host cell and are conserved between Stx2a and Stx2k except for AA16, which is part of site 1, and AA31, which is part of site 2. Figure 3B exhibits the pentameric B-subunit with residues that constitute site 1, which was shown as sticks with carbon, nitrogen, and oxygen in grey, blue, and red, respectively. Also included in Figure 3B is the C-terminal helix of the A-subunit that resides at the centre of the pentameric B-subunits. A superimposed view of the residues of site 1 in Stx2a and Stx2kE167Q is shown in Figure 3C. The residues of binding site 3 in Stx2a and Stx2kE167Q could also be closely superimposed. Overall, all three sites for receptor binding are very similar in Stx2kE167Q and Stx2a, indicating that their specificity for receptors may be very similar. However, differences in site 1 and site 2 (AA16 and AA31) could contribute to specificity for host cell toxicity.

### 3.4. Development of a Polyclonal Antibody and ELISA for Detection of Stx2k

We previously reported the isolation of nine *stx2k*-containing *E. coli* strains, but Stx2k was barely detectable in some strains using a commercial test kit [37]. To develop a sensitive immunoassay for Stx2k, a collection of antibodies against Stx2 were screened and the monoclonal antibody (mAb) Stx2e-3 and a polyclonal antibody (pAb) raised against Stx2e and Stx2a [38,64], respectively, were found to be able to detect the A-subunit of Stx2k, but the signal was weak (Figure 4A). Therefore, a new pAb was developed using Stx2kE167Q toxoid as an antigen. The two rabbits immunized with the Stx2kE167Q recombinant toxoid showed high antibody serum titers (a signal to noise ratio of antisera:preimmune sera was greater than 500:1 in ELISAs coated with 10 ng/mL of Stx2k on the plates and detected with antisera from the 2^nd^ bleeding diluted 1:8,000 in PBS). The binding of the new pAb to purified Stx2k was improved based on results from Western blot (Figure 4A). The new pAb was also capable of detecting the A-subunit of Stx2k present in bacterial culture supernatants by Western blot (Figure 4B). Although the antigen used for antibody production contained both the A- and B-subunits, the pAb Stx2k predominantly bound to the A-subunit in Western blot analyses, which is consistent with previous results [64].

When the pAb Stx2k was used as a capturer and the mAb Stx2e-3 as a detector in a sandwich ELISA, the limit of detection for Stx2k was 190 pg/mL, and a linear standard curve with a R^2^ = 0.990 was observed in the range of 0.19–12.5 ng/mL. However, when the pAb Stx2k was used as the detector and the mAb Stx2e-3 as a capturer, a linear standard curve with a R^2^ = 0.979 was observed in the range of 0.048–12.5 ng/mL, but the limit of detection (LOD) was 48 pg/mL for Stx2k, a four-fold increase in sensitivity (Figure 4C). In contrast, when the pAb Stx2a was used with mAb Stx2e-3, the LOD for Stx2k was 480 pg/mL or 780 pg/mL, respectively, but Stx2k was not detectable when pAb Stx2a was used in conjunction with mAb Stx2a-2 [44] (Appendix A). Therefore, the antibody pair: mAb Stx2e-3 (as a capturer) and pAb Stx2k (as a detector) was specifically used for detection of Stx2k. When the same antibody pair was used for detection of Stx2a, the sensitivity decreased eight-fold (Figure 4D), which was expected as the detection antibody was generated using the Stx2k antigen that differs from Stx2a by 5% at the amino acid level.

Previously, we used the Duopath STEC Rapid Test to detect Stx2k produced by nine *stx2k*-containing strains and no toxin was detectable from the culture supernatant of strain STEC 388. Consistent with this result, a cytotoxicity assay also failed to detect toxicity from supernatant of strain STEC 388 [37], suggesting a very low level of Stx2k was produced by STEC 388. Using the Stx2k toxoid as a standard and the ELISA developed in this study, we were able to detect and quantify the amount of Stx2k toxin produced by both high and low expressing strains, including strain STEC 388. The Stx2k detected in culture supernatants of four STEC strains ranged from 4.5 ng/mL (strain STEC 388) to 485 ng/mL (strain STEC 309) (Figure 4E).

### 3.5. Cytotoxicity of *Stx2k*

Infection by STEC strains with a high expression of Stx is often associated with severe clinical outcomes, however, other virulence factors produced by STEC can also affect the toxicity of a strain [65,66,67,68]. Therefore, we purified Stx2k from the MMC-induced supernatant of STEC 309 (a clinical isolate) [37] to assess the toxicity of Stx2k specifically. Approximately 175 µg of Stx2k was purified from the supernatant of a 1 L overnight culture using an affinity column conjugated with mAb Stx2e-3, followed by gel filtration (Figure 1). The biological activity of the purified Stx2k was assessed by a microtiter cytotoxicity assay using Vero cells (Figure 5A,B). A dose response was observed between 10 fg/mL–100 ng/mL of toxin and a CD_50_ (the toxin concentration that kills 50% of cells) of 0.619 ± 0.16 pg/well was measured for Stx2k. Compared to the CD_50_ of 0.078 ± 0.027 pg/well for Stx2a, Stx2k was eight-fold less toxic.

### 3.6. Neutralization of *Stx2k*

Antibodies raised against Shiga toxin can vary in their neutralizing ability, therefore, the antibodies used in this study were tested for their ability to neutralize Stx2k. Vero cells were incubated with either toxin alone, antibodies alone, or toxin pretreated with antibodies, and compared to the untreated Vero cell control. Treatment with mAb Stx2e-3 did not reduce toxicity of Stx2k compared to toxin alone (Figure 5C) while pAb raised against Stx2a was able to neutralize 22% of the Stx2k toxicity compared to toxin only control (*p* > 0.0001). However, the pAb raised against Stx2k completely neutralized the cytotoxic effect of Stx2k on Vero cells (*p* > 0.0001). The superior neutralizing ability of the pAb Stx2k could be useful for management of infections caused by STEC strains that produce Stx2k.

### 3.7. Receptor Binding

Shiga toxins enter cells through interaction with glycolipid receptors, either Gb3-LPS or Gb4-LPS, on the host cell membrane. The difference in receptor preference between subtypes is thought to play a critical role in host range and cytotoxicity [16,18,69]. The sequence and structure of the receptor-binding sites are very similar between Stx2k and Stx2a, however there is a residue difference in both Site 1 and Site 2 (Figure 3), raising the possibility that variation in Vero cell cytotoxicity is due to differences in receptor preference. Receptor preference of Stx2k was tested by utilizing a “Sandwich” ELISA [39]. Formaldehyde-fixed *E. coli* cells expressing either Gb3 or Gb4 were coated onto 96-well plates and used to capture the toxins [42,43]. The mAb Stx2e-3 was used to detect the bound Shiga toxin. It was found that Stx2k predominantly bound to the Gb3 receptor and not the Gb4 receptor, which is very similar to Stx2a (Figure 6) and suggests that any difference in cytotoxicity between Stx2k and Stx2a may not be due to receptor-binding preference.

### 3.8. Stability of *Stx2k*

Many gastrointestinal diseases result from ingesting foods contaminated with STEC and Stxs. In order to improve the quality and safety of the food supply, information is needed to determine the stability of Stxs. It has been reported that toxin stability varied among subtypes of Stx2 [25,39]. To determine Stx2k stability, both Stx2a and Stx2k were exposed to increasing temperatures (63 °C, 74 °C, and 95 °C) in a water bath. These temperatures were chosen based on previous work in the laboratory and guidelines that were determined by the U.S. Food & Drug Administration to inactivate 90% of the pathogens and toxins present in food samples (https://www.fda.gov/food/guidance-regulation-food-and-dietary-supplements). Stx2k toxicity was completely abrogated at 95 °C, but it still had 40% activity when exposed to 74 °C, as compared to the untreated toxin control (*p* > 0.002). Treating Stx2k at 63 °C for 1 h had no impact on toxin activity (*p* > 0.14) (Figure 7A). To determine the effect of pH on Stx2k, the toxin was subjected to extreme pH conditions (pH 1.5–pH 4) for 1 h at room temperature. After exposure, the toxin was diluted and neutralized with DMEM media before adding to the Vero cells. No decrease in cytotoxicity was observed for Stx2k at pH 3 and pH 4. However, Stx2k was completely inactivated when it was exposed to pH 1.5 and 2 (*p* > 0.00041 and *p* > 0.00045, respectively) (Figure 7B). The effect of buffer without toxin on Vero cells was negligible. No significant difference in thermal stability and low pH tolerance was found between Stx2k and Stx2a.

## 4. Discussion

The aim of this study was to characterize the structure and biological properties of the newly identified Stx2k subtype [37]. To this end we developed a recombinant toxoid with a single amino acid substitution at position 167. The recombinant toxoid was advantageous as it allowed us to purify nontoxic protein at the high concentrations necessary for crystal structure experiments and antibody generation for immunoassays. Analyses of the crystal structures revealed that Stx2k and Stx2a are highly similar, and the Stx2k enzymatic active site is nearly identical to that of Stx2a. However, Stx2k was eight-fold less toxic than Stx2a in Vero cell assays. It is not clear what causes the reduction in cytotoxicity of Stx2k.

Differences in toxicity among Stx subtypes can be attributed to variation in receptor binding, yet there is only a 5% sequence difference in the B-subunits of Stx2k and Stx2a. Accordingly, both toxins predominantly bind to the Gb3 receptor, suggesting that receptor preference is likely not the reason that the toxins differ in potency. The crystal structure revealed that two of the amino acids (AA16 and AA31 in Figure 2C) that differ between Stx2k and Stx2a are surface-exposed residues in receptor binding sites 1 and 2, respectively. All the Stx2 subtypes identified to date have either an asparagine or an aspartic acid at position 16, and an N16D mutation has been shown to increase cytotoxicity 200-fold of the Stx2c subtype [70]. Moreover, it was reported that some Stxs with the same receptor preference displayed differences in receptor-binding kinetics and it was speculated that a slow receptor-toxin dissociation may increase the time allotted for toxin uptake and thus increase toxicity [16]. The receptor preference assay reported here does not take binding kinetics into account and as such we cannot rule out the possibility of receptor-toxin dissociation rates as the reason for the differences in toxin potency. It is possible that the amino acid differences at binding sites 1 and 2 affect receptor-toxin binding kinetics, resulting in the difference in potency between Stx2a and Stx2k.

Despite their differences in cytotoxicity, Stx2a and Stx2k exhibit similar stability profiles. During an infection, orally ingested STEC must initially survive the harsh environment of the stomach and then compete with other gut microorganisms to establish intestinal colonization. Accordingly, Stxs produced in the lumen exhibit a generalized stability at low pH and some Shiga toxins can tolerate as low as pH 2 [25,39]. Both Stx2k and Stx2a are destabilized at a pH of 2 or lower. In addition to pH, Stx subtypes demonstrate variability in heat resistance [39]. *E. coli* strains can be eliminated from food products by heating to 63 °C, a standard temperature used in pasteurization, yet both Stx2k and Stx2a toxins produced by STEC remain fully active at 63 °C and partially active even at 74 °C, suggesting that temperatures used to inactivate *E. coli* are not necessarily effective on toxins. Therefore, illness due to ingesting toxin-contaminated foods could still pose a health problem.

Stx subtypes are highly similar in sequence and structure, and Stx antibodies are often cross-reactive, which makes it possible to apply the same immunological-detection kit to multiple subtypes of Stx2 [40,44]. However, as subtype identity diverges from Stx2a, detection becomes limited and necessitates the development of new antibodies and immunoassays capable of detecting emergent subtypes. The crystal structure of Stx2k revealed that 13 of the 17 amino acid differences between the Stx2k and Stx2a are located on the surface of the toxin. These surface residue differences likely constitute distinct epitopes specific to Stx2k antibodies. Therefore, it was unsurprising that so few Stx2 antibodies in our collection bound and neutralized Stx2k. We previously reported that Stx2k was undetectable in culture supernatant of strain STEC 388 [37]. However, Stx2k was successfully detected using the ELISA with the pAb Stx2k developed here. Moreover, the new Stx2k pAb was able to completely neutralize the cytotoxicity of Stx2k on Vero cells. Taken together, our work demonstrates that development of new antibodies and immunoassays for emerging Stx subtypes could improve efficacy of detection, minimize the incidence of false negatives, and thus reduce outbreaks and disease mismanagement.

Stx2k-producing bacterial strains have been isolated from a variety of sources, including animals, raw meats, and diarrheal patients [7] and the levels of Stx2k produced by different strains varied significantly (~100-fold across the four strains studied here). Although no reports of HUS have been attributed to Stx2k yet, the wide distribution of strains suggests that Stx2k-producing STEC have become well established in environment. Two out of the nine Stx2k-producing strains isolated were from diarrheal patients and possessed other virulence factors such as hemolysin E (hlyE), adhesion (fimH, ehaA), type III secretion system effector (espL, espX), and non-LEE-encoded type III secreted effector (espR). Additionally, one of the asymptomatic strains, STEC 388, phylogenetically clustered closely to clinical STEC/ETEC hybrid strains from a diarrheal patient in Sweden [16] and carried virulence factors of ETEC in addition to Stx2k. The extraordinary genomic plasticity and highly mobile nature of *stx2k-*converting phage creates a high probability of emergent pathogens with hybrid virulence profiles which could cause serious consequences for patients.

## 5. Conclusions

In conclusion, the structure of Stx2k determined at 2.29 Å resolution is highly similar to that of Stx2a, however, Stx2k is distinct from Stx2a, serologically. Stx2k was eight-fold less toxic than Stx2a on Vero cells, but Stx2k and Stx2a share similar properties in receptor-binding preference for Gb3, thermostability, and acid tolerance. Our study suggests that the food safety procedures utilizing heat and pH inactivation of bacteria may not be sufficient for toxins produced by bacteria. To effectively inactivate Stx2a and Stx2k in buffer, the temperature must be set to >74 °C and the pH must be <3. However, toxins may exhibit different sensitivities if present in food or other matrices, such as milk, beef, egg, and soil etc. The polyclonal antibody developed here exhibited a high neutralization activity that may have therapeutic significance. The ELISA established using the new pAb was capable of detecting Stx2k at concentrations as low as 48 pg/mL. The sensitivity and simplicity of the assay makes it a valuable tool for etiologic diagnosis of STEC infections and accessible for use in developing countries. This research reaffirms the need for the development of novel antibodies and immunoassays as new subtypes of Stx continue to emerge.

## Figures and Tables

**Figure 1 microorganisms-08-00004-f001:**
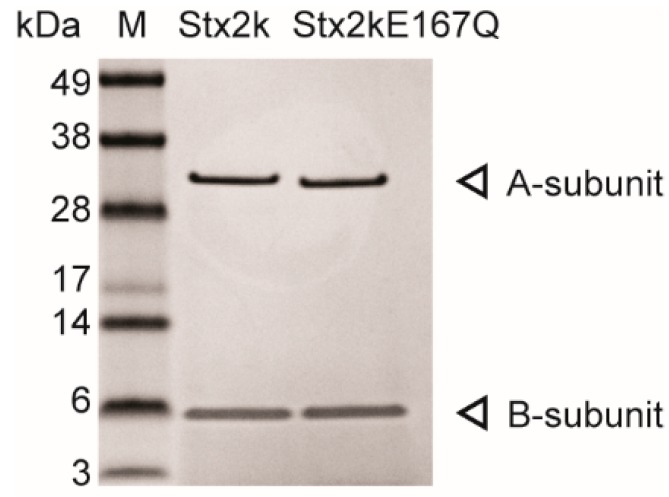
Purified Shiga toxin 2k (Stx2k) and Stx2kE167Q. Stx2k and Stx2kE167Q purified from Shiga toxin-producing *Escherichia coli* (STEC) 309 and BL21-Stx2kE167Q bacterial cells, respectively, were analyzed by Coomassie staining on a 4–12% sodium dodecyl sulfate-polyacrylamide gel electrophoresis (SDS-PAGE). The molecular weights (kDa) of protein markers (M) are indicated next to the marker. The A- and B-subunits are denoted by an arrow at the right side of the panel. Each lane contains 0.5 µg of purified protein.

**Figure 2 microorganisms-08-00004-f002:**
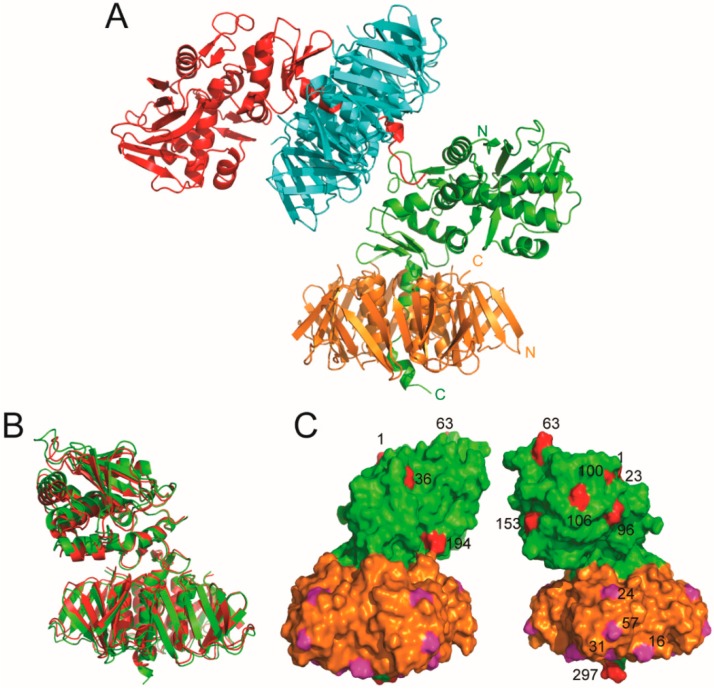
Structure of Stx2kE167Q. (**A**) The unit cell contained two biological units of Stx2kE167Q, each of them is constituted by an A-subunit (shown in red and green, respectively) and five B-subunits (shown in cyan and gold, respectively). The C-terminus of one of the A-subunits (shown in red) is involved in the packing of the two biological units. The N- and C-terminals of one Stx2kE167Q subunits are labeled. (**B**,**C**) Structure comparison of Stx2a and Stx2kE167Q. (**B**) The structure of one of the biological units of Stx2kE167Q (green) is superposed with that of Stx2a (red) and shown as a ribbon diagram. (**C**) A surface presentation of Stx2kE167Q with the A-subunit shown in green, and the B-subunit in gold. Residues in subunit A that are not identical with those in Stx2a is shown in red. Residues in subunit B that are not identical with those in Stx2a are shown in magenta. The non-identical residues in the A-subunit and one of the B-subunits are numbered.

**Figure 3 microorganisms-08-00004-f003:**
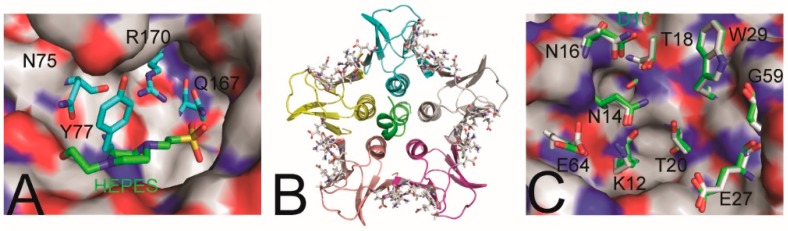
Active center and Site 1 of receptor binding of Stx2kE167Q. (**A**) The configuration of the active center residues is shown in a stick representation. Also shown is a HEPES molecule labeled in green, which resides in the active center cavity of the A-subunit of the toxoid. (**B**) The distribution of receptor-binding site 1 in the pentameric B-subunit of Stx2kE167Q. Five of the subunits are shown in different colors. The C-terminal helix of the A-subunit is shown in green at the center of the pentamer. Most of the protein shown is presented in a ribbon diagram. The residues that constitute the receptor-binding site 1 are shown in a stick presentation. (**C**) Superposition of the residues of site 1 in Stx2a and Stx2kE167Q. Carbons in the amino acid residues of Stx2kE167Q and Stx2a are shown in grey and green, respectively. Nitrogen and oxygen are shown in blue and red, respectively. One representative site 1 is shown in a stick presentation on the surface of the rest of the protein. The AA16 of Stx2a is labeled in green.

**Figure 4 microorganisms-08-00004-f004:**
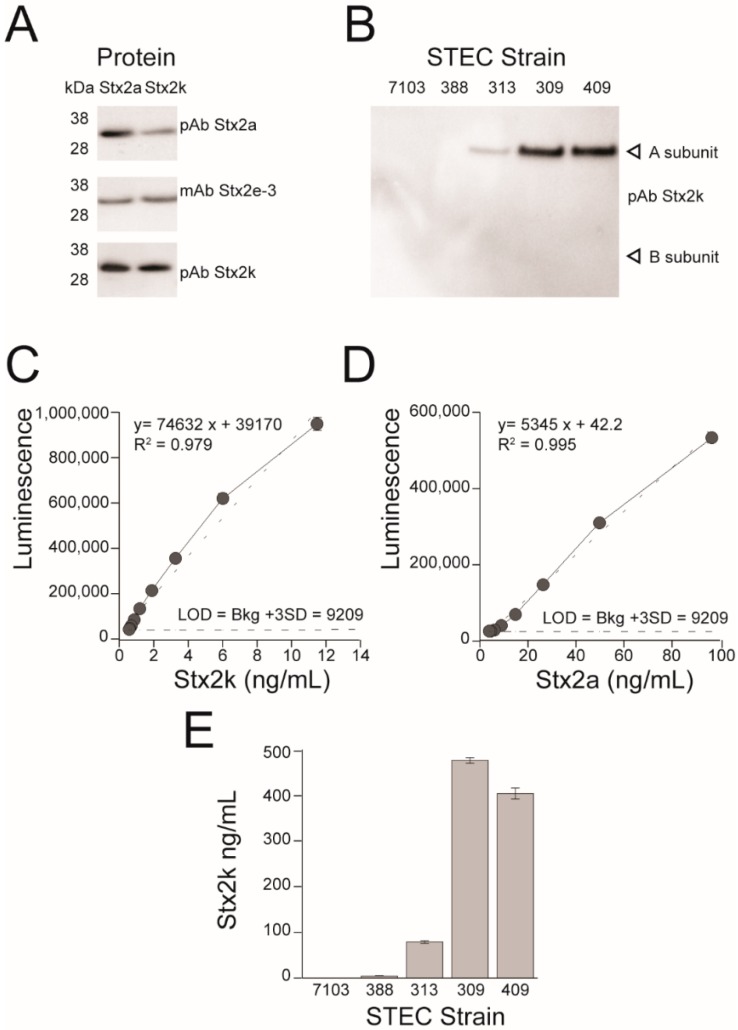
Detection and quantification of Stx2k. (**A**) Western Blot of purified Stx2k and Stx2a following sodium dodecyl sulfate-polyacrylamide gel electrophoresis (SDS-PAGE). Samples: 25 ng of protein were used when probed with either polyclonal antibody (pAb) Stx2a or pAb Stx2k, and 50 ng were used when probed with monoclonal antibody (mAb) Stx2e-3. (**B**) Western Blot of 10 µL mitomycin C (MMC)-induced bacterial culture supernatants and probed with pAb Stx2k. Strains were induced with MMC at Optical Density of 0.6 at the wavelength 600nm (OD_600_) 0.9, grown overnight at 37 °C and harvested at OD_600_ 6.0. STEC 7103 is a *stx− E. coli* strain. STEC 388, 313, 309, and 409 are *stx2k* expressing strains. (**C**) Standard curve of Stx2kE167Q. Detection of Stx2kE167Q in phosphate-buffered saline (PBS) using mAb Stx2e-3 as a capture antibody and pAb Stx2k as a detection antibody. Black circles represent the mean of triplicate readings ± one standard deviation (SD). Horizontal dashed line equals the mean of triplicate readings from PBS control plus three SD. (**D**) Standard curve of Stx2aE167Q. Detection of Stx2aE167Q in PBS using mAb Stx2e-3 as a capture antibody and pAb Stx2k as a detection antibody. Black circles represent the mean of triplicate readings ± one SD. Horizontal dashed line is the mean of triplicate readings from PBS control plus three SD. (**E**) Detection of Stx2k produced by STEC strains. Stx2k concentrations in MMC-induced culture supernatants were estimated using the standard curve plotted in Figure 4C.

**Figure 5 microorganisms-08-00004-f005:**
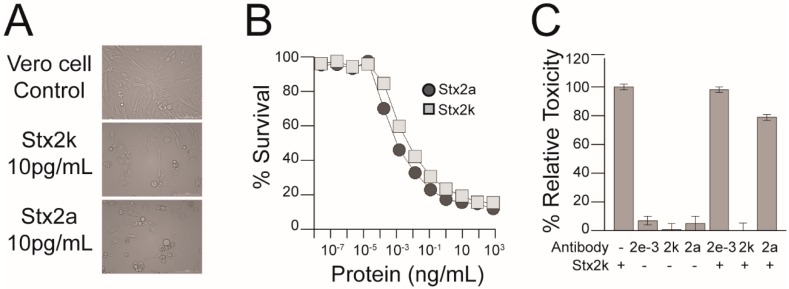
Toxicity and neutralization of Stx2k. (**A**) Representative images (20× magnification) of Vero cells intoxicated with 10 pg/mL of Stx2k, Stx2a, or media control. (**B**) A dose response curve of Stx2k- and Stx2a-treated Vero cells. Vero cells were treated with 10-fold serial dilutions of toxins from 1 µg/mL to 0.01 fg/mL in Dulbecco’s Modified Eagle Medium (DMEM) for 48 h at 37 °C with 5% CO_2_. Cell viability was analyzed with CELLTITER-GLO 2.0 ASSAY and luminescence was measured on a Victor3 plate reader. The percent survival is calculated as (the counts per second (CPS) of the experimental well/the average CPS of the DMEM control) × 100%. Dark grey circles and light grey squares represent the mean of three determinations ± one SD from samples treated with Stx2a and Stx2k, respectively. (**C**) Neutralization of Stx2k cytotoxicity by antibodies. Vero cells were treated with Stx2k (5 ng/mL), the indicated antibody (10 µg/mL), or Stx2k that was preincubated with one of the indicated antibodies. Vero cells were treated for 48 h at 37 °C with CO_2_ and the percent relative toxicity is calculated relative to the toxin only condition. Each experiment was repeated three times.

**Figure 6 microorganisms-08-00004-f006:**
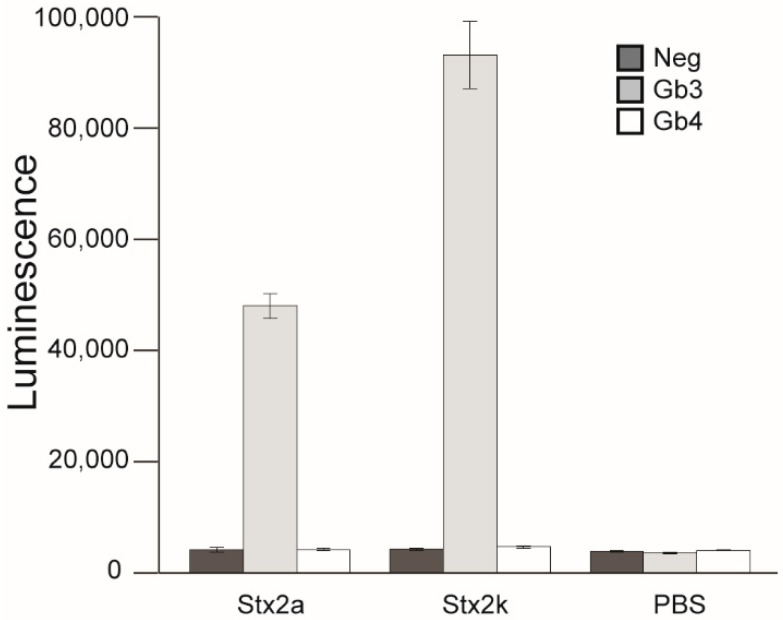
Receptor-binding preference of Stx2k. Binding was measured in an ELISA using *E. coli* cells expressing Gb3-LPS (Gb3 mimic) or Gb4-LPS (Gb4 mimic), or control cells. Purified Stx2a and Stx2k (100 µL of 50 ng/mL) were added to wells coated with bacterial cells expressing Gb3 or Gb4 receptors. mAb Stx2e-3 was used to react with bound toxins. Goat anti-mouse horse radish peroxide (HRP) was used for detection. Luminescence was measured on a Victor3 plate reader. Data shown represent the mean of triplicate luminescence counts ± one SD from one representative experiment. Each experiment was repeated four times.

**Figure 7 microorganisms-08-00004-f007:**
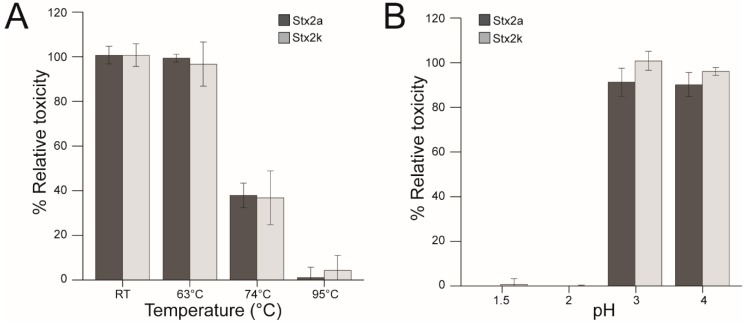
Stability of Stx2k and Stx2a. (**A**) Thermal stability of Stx. Toxins were heat treated in a water bath for 1 h at the indicated temperatures, cooled on ice, and then diluted to 5 ng/mL in pre-warmed Dulbecco’s Modified Eagle Medium (DMEM) before use in Vero cell assays. Percent relative toxicity is calculated relative to toxins treated at room temperature (as 100%). Dark grey bars are Stx2a. Light grey bars are Stx2k. (**B**) pH stability of Stx. 500 ng/mL of toxins were treated for 1 h in 250 mM sodium acetate buffer at the indicated pH and then neutralized and diluted to 5 ng/mL in DMEM before use in the Vero cell assay. Percent relative toxicity is calculated relative to toxins in DMEM (as 100%). Dark grey bars are Stx2a. Light grey bars are Stx2k. Data shown represent the mean of triplicate readings ± one SD from one representative experiment. Each experiment was repeated three times.

**Table 1 microorganisms-08-00004-t001:** Strains used in this study.

Strain Name	Protein Expressed	Reference
BL21-*stx2kE167Q* *	Stx2kE167Q	This study
RM10638 *	Stx2a	[40]
RM7103	n/a	[41]
STEC309 *	Stx2k	[37]
STEC313	Stx2k	[37]
STEC388 ^a^	Stx2k	[7,37]
STEC409	Stx2k, Stx2e	[37]
CWG308 pJCP-Gb3	Gb3-LPS	[42]
CWG308 pJCP-*lgt*CDE	Gb4-LPS	[43]
CWG308	n/a	[42]

An * indicates the strains were used for protein purification. ^a^
*stx2k* gene in strain STEC388 was initially assigned as a Stx2e variant [7], but further analysis reassigned it as a new Stx2k subtype [37].

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
