# Peer review of "Structural and Functional Characterization of Stx2k, a New Subtype of Shiga Toxin 2"

_microorganisms, 2019, doi:10.3390/microorganisms8010004_

Round 1

Reviewer 1 Report

This is a revised submission on the characterization of Stx2k and the purification and structure of Stx2k toxoid. The errors in figure legends and figure order have been fixed, and the other comments have been addressed. 

Comments (line #s refer to "clean" copy of manuscript):

lines 41-43 - STEC have been known to be human pathogens since 1982 when the first outbreak occurred in the US. STEC were not really identified before that time. These lines need to be re-written. Perhaps the authors are referring to STEC isolates from animals that are not yet known to be human pathogens?

line 86 - strongly suggest deleting "result in serious outbreaks" - there is no real support for the serious outbreak portion of this sentence. 

line 131 - please provide a reference for mAb Stx2-1 or provide a brief description of it (which subunit does it bind to?).

line 236.  What restrictions on generating biologically active toxins using recombinant technology are the authors referring to?  The stxs are no longer select agents. The only restriction this reviewer is aware of is that an intact stx operon may not be introduced into a wild type STEC. Suggest deleting this portion. 

lines 238-240 - Sentence is awkward. one possible fix:  The  E167Q mutation was chosen because E167 is known to be the active site residue for known ribosomal inactivating proteins (RIPs), including the Stxs, and because the E167Q mutation does not alter the structure of RIPs [refs].

Reviewer 2 Report

The authors have adequately addressed my prior technical comments. I don't have any other concerns. This paper will make an important addition to the STEC field.

Author Response

Thank you. 

This manuscript is a resubmission of an earlier submission. The following is a list of the peer review reports and author responses from that submission.

Round 1

Reviewer 1 Report

The manuscript by Hughes et al described structure function of a new Stx2 variant - Stx2k. Description of such a new Stx2 variant would be of high interest to the microbiology community. The structural data is strong, and obtaining a resolution of 2.29A is a clear strength of the paper. However, my enthusiasm is substantially reduced by what I perceive to be major technical issues. In particular, the paper describing Stx2k production (ref. 37) is not published or available - therefore, it is not possible to determine whether Stx2k is indeed a new Stx2 variant. Additionally, the methodology for protein purification and the antibody generated against Stx2k raises some doubts in my mind about cross-reactivity with Stx2a. If these key concerns can be addressed, I would be supportive.

1. My main concern is that the paper describing the identification of Stx2k (ref. 37) is not yet published and is described as in preparation. The current manuscript should be delayed until ref.37 is peer-reviewed and accepted. Without publication of ref. 37 or inclusion of the data from that paper here, it is not possible to judge whether Stx2k is indeed a new variant of Stx2, and what the relevance of this new variant is to STEC toxicosis.

2. In the methods, cloning of Stx2k refers to Genebank sequence AM904726 and ref. 26. Genebank sequence AM904726 is for Escherichia coli Stx2a. If the protein made was based on this sequence, it is not Stx2k.

3. With regards to the above point, how was full length Stx2k cloned into the His-vector? What is the order of A and B subunits?

4. Strains of STEC used (Table 1). There are multiple citations to ref.37, which is not available. Additionally, there is a citation to ref.8 as producing Stx2k. I could not find any mention of Stx2k in ref.8.

5. Purification of wild-type Stx2k: It is unclear which strains were used to purify Stx2k and Stx2a, and whether the purification method was specific - i.e. method used for Stx2a did not purify Stx2k. In this regard, Fig.7A shows that the mAb against Stx2e equally binds Stx2a and Stx2k.

6. Sequence alignment: In addition to the A-subunit, it will be important to show alignment of the B-subunits, because differences in the sequence of the B-subunit between Stx1 and Stx2 are critical in mediating retrograde transport after endocytosis.

7. Stx2k antibody: In Fig.7, the anti-Stx2k antibody is detecting Stx2a just as well. This raises doubts  about which protein is being detected in the STEC strains. Extending from this, it raises concerns about the cell-based assays performed subsequently.

Reviewer 2 Report

This manuscript describes the purification, characterization, and crystal structure of an apparent toxoid of Stx2k. Overall they appear to have successfully purified both the toxin and toxoid, and provide the crystal structure of the toxoid. They also describe the interactions between various antibodies and the toxin or toxoid.  They reference to a paper that describes the identification of Stx2k, but that unpublished work was not provided. It would have been helpful to see the manuscript of the original identification of Stx2k. 

The abstract should be changed to indicate that they determined the crystal structure of the toxoid rather than the toxin -  and anywhere else in the manuscript that same correction should be made. The manuscript is quite long, and there are some mistakes with an incorrect Figure legend (#2) and figures switched (9&10) as well as some other mistakes, noted below. Figure 3 should be eliminated since that information would presumably be in reference #37 (since they used reference #37 to indicate the level of similarity between Stx2a and Stx2k).

The figure legend to Figure 2 is incorrect, and the figures in figure 2 could be combined into Figure 4. Table 2 could go into supplementary since the crystal structure is not really the focus of the paper overall.

lines 314-317 --it looks like there is a difference for site 2 between Stx2a and Stx2k?  an S as compared to an N as it appears in Figure 3. Unclear why there is no discussion of this?  

It seems quite odd to have the cytotoxicity not shown until figure 8--is that because the evidence of cytotoxicity is already shown in the unpublished reference 37?  Why wasn't it shown that the E167 mutation eliminated toxicity - that could be stated in the methods even, as not shown. 

line 232 suggest adding "large quantities" in front of Stx2k--because clearly Stx2k was purified by the authors--the difficulty is in getting enough, not getting it at all.  

line 235 --should add in Stx2e --"was shown to destroy the toxicity of Stx2e, but keep..." 

line 273 - "the structures are similar" -does that refer to figure 4? 

lines 295-297 - please rewrite for clarity.

Figure 5 - why showing two panels of the same thing? --there is no reference to 2a in the figure legend, so presumably those are both Stx2k since the Q167 is labeled. (but line 298 suggests Figure 5 is a superimposition- which it cannot be since there is no E167 for the Stx2a) - this section is unclear.  

lines 410-413 It is not clear that the assertion that a highly neutralizing antibody is required for each Stx. A monoclonal to Stx2a neutralizes in vitro to about 67% but completely protects animals from Stx2a in multiple assays. (Infect Immun. 2009 Jul;77(7):2730-40, Clin Vaccine Immunol. 2015 Apr;22(4):448-55.)

Supplementary figure 1 is unnecessary. 

Other comments

Lines 230 -235. Line 230 indicates that nothing is known about Stx2k in structure or function, but line 235 indicates that you knew the exact active site residue to mutate. This seems contradictory. Please rewrite line 230 or delete and reword the next line. 

Figure 8 - something is wrong with Panel C--maybe just mislabeled. Panel C only shows neutralization of Stx2k, but the text in the legend talks about neutralization of Stx2a.

minor comments

Lines 190 and 195 - do not capitalize toxins on those lines. 

There is a symbol in lines 174 and 215 that is incorrect.

Line 261 C-terminus rather than C-terminal.

line 322 data rather than date

If Figure 3 is kept, please label the figure panels A and B for clarity.